EMBO
Molecular Medicine

# Succession of transiently active tumor-initiating cell clones in human pancreatic cancer xenografts

Claudia R Ball[1,†], Felix Oppel[1,†], Karl Roland Ehrenberg[1,2,†], Taronish D Dubash[1], Sebastian M Dieter[1,3], Christopher M Hoffmann[1], Ulrich Abel[1], Friederike Herbst[1], Moritz Koch[4,5], Jens Werner[4,6], Frank Bergmann[7], Naveed Ishaque[8,9], Manfred Schmidt[1,3], Christof von Kalle[1,3], Claudia Scholl[1], Stefan Fröhling[1,3,10], Benedikt Brors[8], Wilko Weichert[3,7], Jürgen Weitz[4,5] & Hanno Glimm[1,3,*]

## Abstract

Although tumor-initiating cell (TIC) self-renewal has been postulated to be essential in progression and metastasis formation of human pancreatic adenocarcinoma (PDAC), clonal dynamics of TICs within PDAC tumors are yet unknown. Here, we show that long-term progression of PDAC in serial xenotransplantation is driven by a succession of transiently active TICs producing tumor cells in temporally restricted bursts. Clonal tracking of individual, genetically marked TICs revealed that individual tumors are generated by distinct sets of TICs with very little overlap between subsequent xenograft generations. An unexpected functional and phenotypic plasticity of pancreatic TICs *in vivo* underlies the recruitment of inactive TIC clones in serial xenografts. The observed clonal succession of TIC activity in serial xenotransplantation is in stark contrast to the continuous activity of limited numbers of self-renewing TICs within a fixed cellular hierarchy observed in other epithelial cancers and emphasizes the need to target TIC activation, rather than a fixed TIC population, in PDAC.

**Keywords** clonal dynamics; pancreatic cancer; phenotypic plasticity; tumor-initiating cells
**Subject Categories** Cancer; Digestive System; Stem Cells

## Introduction

Pancreatic ductal adenocarcinoma (PDAC) is one of the deadliest human cancers due to frequent local recurrence and metastases (Warshaw & Fernandez-del Castillo, 1992; Hidalgo, 2010; Siegel *et al*, 2013). Most patients die within 1 year after diagnosis and only up to 6% will survive more than 5 years (Ferlay *et al*, 2010; Jemal *et al*, 2010). Quantification of the genetic evolution of passenger mutations within PDAC primary tumors and metastases has indicated a natural history of at least a decade of PDAC tumors before metastatic spread (Yachida *et al*, 2010). Still, around 80% of all patients suffer from locally advanced or metastasized disease already at the time of diagnosis. Despite recent improvements in clinical response rate and survival time by combining multiple chemotherapeutic drugs, treatment of metastatic disease is not curative, and in most cases, responses last only for a few months (Conroy *et al*, 2011).

Xenotransplantation of patient cancer samples into severely immune-deficient mice is widely used to investigate the biology of human cancer cells *in vivo* due to the lack of alternate experimental *in vivo* models. Disruption of the original tumor's architecture, the transplantation procedure, and the xenogenic environment may thereby influence the behavior of the assayed tumor cell population (Rycaj & Tang, 2015). Despite these limitations, during the last decade xenotransplantation experiments of purified cancer cells in immunodeficient mice have provided compelling evidence for a hierarchical cellular organization within many types of human leukemias and solid cancers. Cancer cells that are exclusively able to regenerate tumors under these conditions have operationally been called tumor-initiating cells (TICs) or tumor stem cells. Tumor

1 Department of Translational Oncology, National Center for Tumor Diseases (NCT) and German Cancer Research Center (DKFZ), Heidelberg, Germany
2 Department of Medical Oncology, National Center for Tumor Diseases (NCT), Heidelberg, Germany
3 German Cancer Consortium (DKTK), University of Heidelberg, Heidelberg, Germany
4 Department of General Surgery, University of Heidelberg, Heidelberg, Germany
5 Department of Visceral, Thoracic and Vascular Surgery, University Hospital Dresden, Dresden, Germany
6 Department of Surgery, University of Munich, Munich, Germany
7 Institute of Pathology, University Hospital Heidelberg, Heidelberg, Germany
8 Division of Theoretical Bioinformatics, German Cancer Research Center (DKFZ), Heidelberg, Germany
9 Heidelberg Center for Personalized Oncology, DKFZ-HIPO, DKFZ, Heidelberg, Germany
10 Heidelberg University Hospital, Heidelberg, Germany
*Corresponding author. Tel: +49 6221 566979; E-mail: hanno.glimm@nct-heidelberg.de
†These authors contributed equally to this work

stem cells have been shown to drive disease progression and metastasis formation in *in vivo* models of a variety of solid cancers, including human pancreatic cancer (Hermann *et al*, 2007; Li *et al*, 2007). In many studies, expression of phenotypic markers such as the cell surface glycoproteins CD133 and CD44 has been used to separate TICs from the bulk of tumor cells that lack tumor regenerative activity (Al-Hajj *et al*, 2003; Ricci-Vitiani *et al*, 2007). Nevertheless, the functional relevance of these markers for achieving or maintaining a stemness state remains unclear, and recent studies questioned a strict association of TIC function with a marker-defined phenotype (Magee *et al*, 2012).

According to the cancer stem cell model, self-renewal activity is an exclusive functional capability of stem cell-like cells within tumors and required for long-term tumor maintenance and progression (Reya *et al*, 2001; Dick, 2008; Nguyen *et al*, 2012). In line with this, we and others have previously identified extensively self-renewing long-term TIC at the top of a hierarchically organized TIC compartment in human colorectal cancer using genetic clonal marking of human cancer cells in serial xenografts (Dieter *et al*, 2011; Kreso *et al*, 2013). Interestingly, the cancer cells that generate primary xenograft tumors are not a homogeneous population, but instead differ in self-renewal and metastasis formation. Extensively self-renewing long-term (LT-) TICs give rise to tumor transient amplifying cells (T-TAC) that contribute to tumor formation only transiently but lack any detectable self-renewal activity. Very rare delayed contributing (DC)-TICs do not contribute quantitatively to tumor formation in primary transplanted mice, but can be recruited to tumor formation upon serial transplantation (Dieter *et al*, 2011).

In this study, we now asked whether a similar clonal organization drives the malignant growth of PDAC in serial xenotransplantation and whether the high aggressiveness of PDAC is reflected in a particularly large fraction of self-renewing TICs. To monitor the clonal dynamics and self-renewal activity of individual self-renewing PDAC TICs during tumor formation *in vivo*, we used a genetic labeling strategy in a serial xenotransplantation model. The results presented here provide direct evidence that a succession of transiently active TICs generating tumor cells in temporally restricted bursts drives long-term progression of serially transplanted PDAC by functional and phenotypic plasticity of PDAC TICs *in vivo*.

## Results

### Efficient expansion of primary human pancreatic TICs

To monitor the clonal dynamics of PDAC TICs *in vivo*, we used lentiviral marking and subsequent highly sensitive integration site

sequencing as previously employed to visualize the clonal dynamics within the colon cancer TIC compartment (Dieter *et al*, 2011). To efficiently mark TICs and limit genetic marking of non-neoplastic cells, we established a functional purification and enrichment strategy of PDAC TICs from primary patient specimens (Table 1, Appendix Fig S1A). Patient-derived PDAC samples were first xenografted in immune-deficient NOD.Cg-Prkdc$^{scid}$Il2rg$^{tm1Wjl}$/SzJ (NSG) mice to deplete human non-neoplastic cells (Fig 1A). Next, PDAC cells from these xenografts were purified and cultured under serum-free conditions in the presence of cytokines supporting the growth of floating tumor spheroids (Hermann *et al*, 2007; Dieter *et al*, 2011; Fig 1B and C). Since non-adherent PDAC spheroids failed to expand or were lost over time, we developed semi-adherent cultures of three-dimensional PDAC colonies with tight cell-to-cell contacts (Fig 1B–F). Here, dissociated PDAC TICs were maintained or expanded, while murine non-neoplastic cells were depleted. Cell surface glycoproteins associated with TIC activity such as CD133, CD44, and CD24 (Al-Hajj *et al*, 2003; Singh *et al*, 2003; Hermann *et al*, 2007; Li *et al*, 2007; O'Brien *et al*, 2007; Ricci-Vitiani *et al*, 2007) were expressed on a variable proportion of cultured PDAC cells. Also, colonies expressed pancreatic acinar (PTF1a, amylase, trypsin) and ductal markers (KRT7; Appendix Table S1) and reliably formed subcutaneous xenografts that faithfully recapitulated the original human tumor histology (Fig 1E and F).

### Molecular marking of individual TIC clones

To determine the clonal kinetics of individual PDAC TICs *in vivo*, we used molecular marking by integrating lentiviral vectors before xenotransplantation (Fig 2A). The unique fusion sequence of the proviral vector and genomic DNA at the integration site (IS) of each transduced cell is inherited by all daughter cells, allowing unequivocal identification of the clonal contribution of individual TICs to tumor formation. Unambiguousness of the clonal marker is not affected by genetic or chromosomal instability within tumor cells as the likelihood of disrupting this exact fusion at base pair resolution is neglectable. IS in established tumors were identified by highly sensitive linear amplification-mediated (LAM)-PCR and subsequent high-throughput sequencing, allowing to detect one IS copy within 1 μg DNA (Schmidt *et al*, 2001, 2007). PDAC colonies derived from three patients (P1–P3) were passaged 1–5 times to deplete non-neoplastic cells, dissociated, and transduced with a self-inactivating lentiviral eGFP vector. Twelve to twenty-four hours after transduction, transduced cells were transplanted subcutaneously, under the kidney capsule or orthotopically into the pancreas of NOD.Cg-Prkdc$^{scid}$Il2rg$^{tm1Wjl}$/SzJ (NSG) mice. Tumors were harvested after 4–16 weeks, and equal proportions of dissociated cells were serially

**Table 1. Patient characteristics.**

| Patient number | Age (years) | Sex | Pathology | Origin in patient | Tumor stage (AJCC) |
|---|---|---|---|---|---|
| P1 | 61 | Male | PDAC | Pancreatic head | IIB |
| P2 | 70 | Female | PDAC | Pancreatic head | IIB |
| P3 | 74 | Male | PDAC | Pancreatic head | IV |
| P4 | 52 | Female | PDAC | Peritoneum | IV |

PDAC, pancreatic ductal adenocarcinoma; AJCC, American Joint Committee on Cancer.

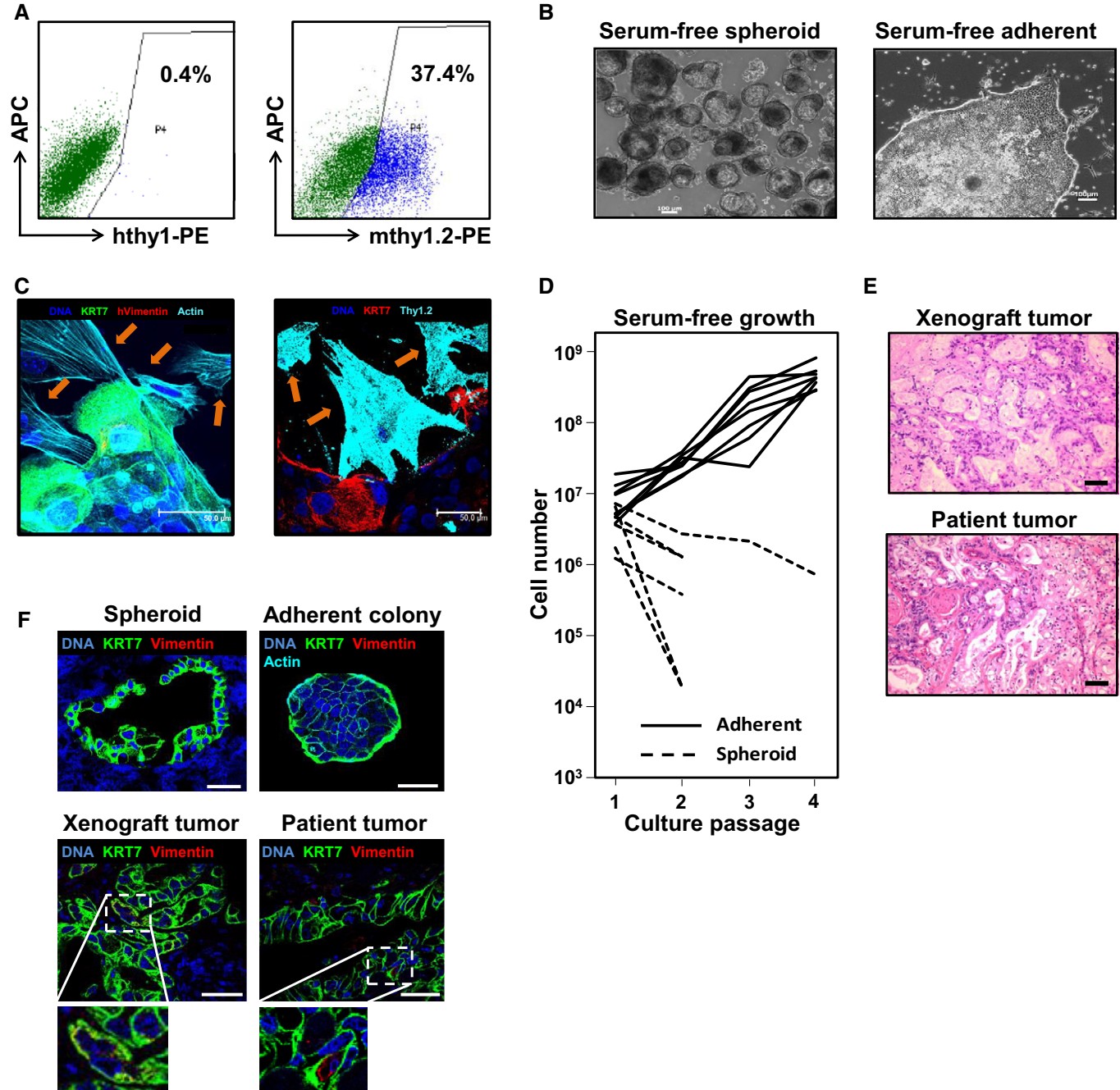

**Figure 1.  Primary pancreatic tumor-initiating cells (TICs) can be expanded in adherent serum-free culture conditions.**

A   Flow cytometry analysis shows absence of human fibroblast marker thy1 epitopes and presence of murine thy1.2-positive fibroblasts in xenograft tumors. Experiments performed in duplicate for patients P1–P4. Representative data from P1 are shown.

B   Primary human pancreatic cells form spheroids in suspension culture, or three-dimensional colonies when grown in adherent culture conditions; scale bars = 100 μm.

C   Indirect immunofluorescence of adherent outgrowth cultures ($n$ = 5) from representative tumors (P1) for human KRT7, human vimentin, murine thy1.2, and species-unspecific actin demonstrates that cells with stromal morphology (arrows) do not express human markers but murine thy1.2. Scale bars = 50 μm.

D   In contrast to spheroid cultures ($n$ = 6), adherent cultures ($n$ = 8) show exponential growth allowing their efficient expansion *in vitro*.

E   Adherent cultures form xenograft tumors reflecting the respective patient tumors histology; scale bars = 100 μm.

F   Spheroids, adherent colonies, and xenografts express high levels of KRT7 comparable to the original patient tumor (patient P1). Only very rare interspersed KRT7+vimentin+ cells are detectable; scale bars = 50 μm. Representative data from patient P1 are shown.

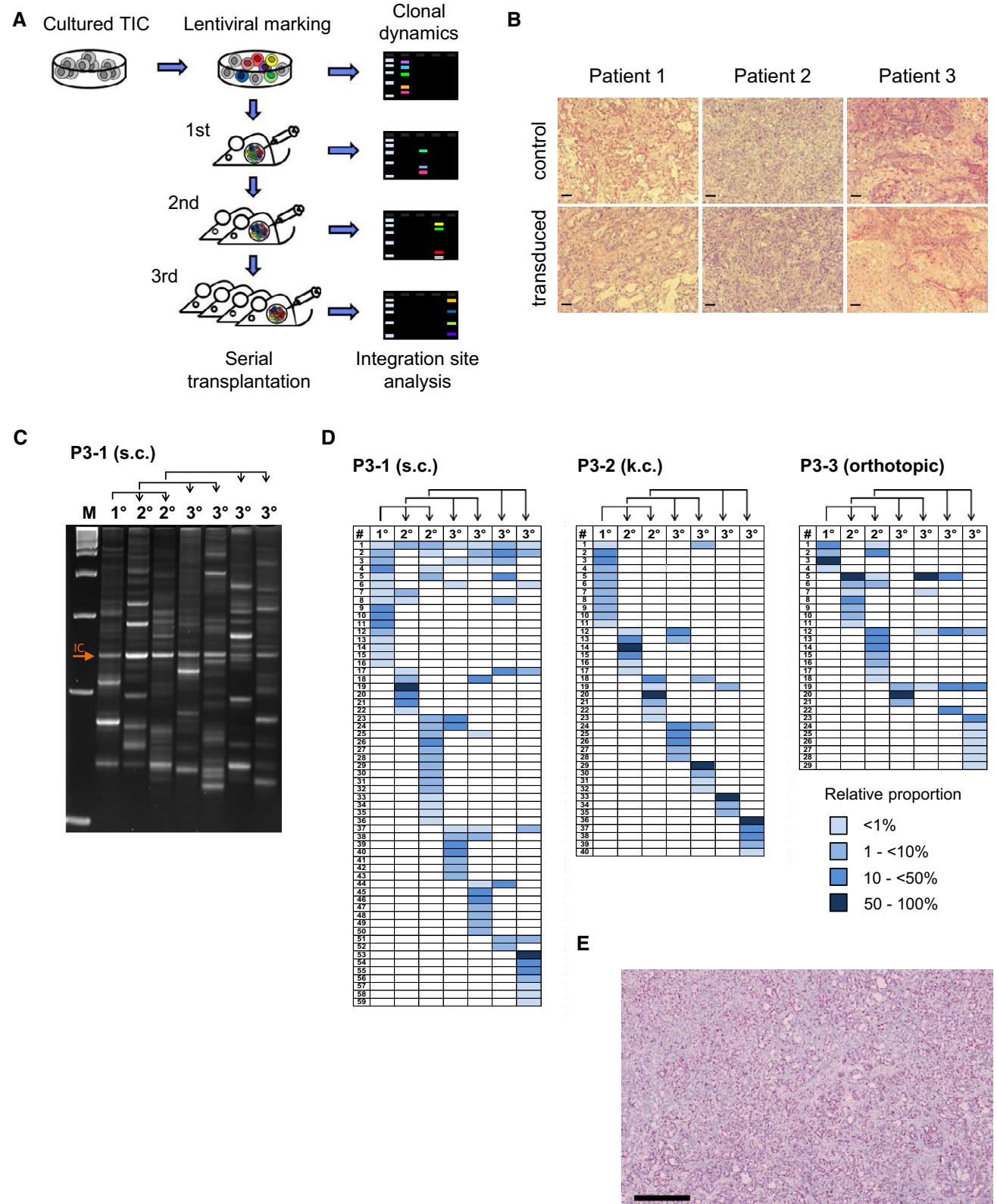

**Figure 2.**

transplanted or used for molecular analyses (Fig 2A). Transduced tumors did not differ in histology (Fig 2B) or growth kinetics compared to untransduced control tumors derived from the same

patients. Three to thirty percent of dissociated cells were used for lentiviral IS analysis, enabling very sensitive detection of clones that had actively divided during tumor formation, whereas mitotically

◀

**Figure 2.  Long-term tumor growth is maintained by transient activity of pancreatic TICs.**

A  Experimental strategy: Tumor-initiating cells (TICs) from three individual primary pancreatic patient tumors were amplified in xenografts, enriched in adherent cultures, genetically marked by lentiviral transduction, and injected subcutaneously, under the kidney capsule, or orthotopically into NSG mice. Primary tumors (1°) were serially transplanted into secondary (2°) and tertiary (3°) recipient mice. Genomic lentiviral integration sites were monitored by LAM-PCR and sequenced to decipher clonal dynamics of long-term pancreatic tumor growth *in vivo*.

B  Lentiviral marking does not affect the histology of xenograft tumors. Scale bars = 100 μm; control = tumors from untransduced cells.

C  Multiple marked cell clones contributed to tumor formation in individual mice during serial transplantation. Each gel band represents an individual integration site amplified by highly sensitive LAM-PCR. Representative data from patient P3 are shown.

D  High-throughput integration site sequencing demonstrated distinct sets of marked clones forming serial tumors within individual experiments. Pairs of mice transplanted with cells from the same donor showed sparse clonal overlap as demonstrated by very low representation of shared integration sites. Clonal analysis was performed for three patients; representative data from patient P3 are shown. Blue fields visualize relative contribution of individual clones as indicated in color legend; rows indicate individual lentiviral integration sites, and columns indicate individual xenografted tumors. Arrows indicate serial transplantation steps; IC: internal control; P3-1/2/3: patient number 3-experiment number 1/2/3. s.c.: subcutaneous transplantation; k.c.: kidney capsule transplantation.

E  Ki67 staining of xenografts derived from TIC cultures demonstrated homogeneous distribution of active cycling and non-cycling cells. Experiments done for P1–P3; representative data from P1 are shown. Scale bar: 500 μm.

inactive marked cells or very small clones would be sampled for IS analysis only randomly.

## Long-term PDAC tumor growth is driven by clonal succession

In total, 236 different IS were detected in xenograft tumors derived from three patients in five independent serial transplantation experiments (P1: 51 IS, P2: 57 IS, P3: 128 IS; Figs 2C and D, and EV1, and Dataset EV1). Genomic IS distribution did not change in serial transplantation and closely reflected the typical spatial pattern of lentiviral integrations (Cartier *et al*, 2009), and we found no evidence for vector-driven clonal expansion. Remarkably, only 0.003–0.113% of initially transplanted transduced cells contributed quantitatively to primary xenograft tumors. In individual primary xenografts, 4–16 unique vector IS were detected, indicating the contribution of multiple marked TICs to tumor formation. Surprisingly, distinct sets of marked clones formed serial tumors derived from the primary xenografts, as IS patterns in secondary and tertiary tumors were strikingly different despite a largely unchanged absolute number of active clones in subsequent tumor generations (Fig 2D). In multiple independent experiments, clones contributing to primary xenograft formation accounted only for 14–39% of clones detected in serial tumors within the respective serial transplantation experiment. The majority of IS was detected at later time points following secondary (103 IS, 2–18 per tumor) or tertiary transplantation (185 IS, 3–21 per tumor). From a total of 236 distinct IS, 183 (78%) were exclusively detected in primary, secondary, or tertiary recipient mice. Even though multiple xenografts were generated within each transplantation generation, 49% of all cell clones were only detected once (primary tumors: 59%; secondary: 50%; tertiary: 45%). Even pairs or triplets of secondary or tertiary tumors showed very little overlap (0–13%) of contributing cell clones (Fig 2D). Despite some inter-patient variability, the majority of IS in all experiments was detected only in single xenografts irrespectively of the generation in which they were active. Strikingly, 75% of clones that contributed strongly to tumor formation in a given generation (≥ 50% of all sequenced reads) were not detected in other xenograft generations indicating that active TIC clones largely generated progeny that lacked TIC activity possibly due to proliferative exhaustion. Thus, molecular marking demonstrates that serial tumor formation in PDAC is mainly driven by TIC clones not actively contributing to tumor formation in earlier but recruited to tumor formation in later generations. Ki67 staining demonstrated very homogeneous

distribution of proliferative active and inactive tumor cells in xenografts derived from all three patients, strongly arguing against major contribution of spatial distribution on TIC activation (Fig 2E). To evaluate whether the observed clonal dynamics are recapitulated *in vitro*, we analyzed clonal kinetics of serially passaged genetically marked primary TIC cultures and *ex vivo*-cultured marked xenograft cells (Fig EV2). Strikingly, every culture passage was formed by a distinct set of actively proliferating cell clones without any significant overlap between the passages, very similar to the clonal dynamics observed in serially passaged xenografts indicating that clonal succession of TIC activity in PDAC is not dependent on the cellular context in tumors *in vivo*.

## Clonal succession is not driven by genetic instability

To understand whether newly acquired genetic alterations contributed to the observed clonal dynamics *in vivo*, we sequenced all coding genes of two primary cultures and derived genetically marked serial xenografts (European Genome-phenome Archive, accession number EGAS00001000882). The mutational landscape of PDAC was remarkably stable during serial xenotransplantation with only few acquired additional mutations (Table 2; Appendix Table S2). In serial xenografts from patient P1, four additional mutations within coding regions of the genes *TTC13, OR4K15, SSPO,* and *TPGS1* were detected with allele frequencies ranging from 2 to 27%. In patient P3 xenografts, only one new mutation in the gene *C10orf12* was detected with a maximum altered allele frequency of 17%. None of these acquired mutations occurred in known cancer driver genes, and all affected genes have been only sparsely found in large-scale cancer sequencing (Forbes *et al*, 2017). We therefore conclude that changes in the functional state of pancreatic cancer cells, and not genetic mutations, underlie the transient activation of TIC clones.

## Mathematic modeling of clone kinetics

Mathematical modeling of *in vivo* cell proliferation within clones was done using a stochastic process, more specifically a linear birth process assuming a homogeneous Poisson process with identical division rates of each single cell within a clone. Analyses were based on confidence interval *P*-values, confidence intervals for the parameter $n$ of a binomial distribution B(n,p), confidence rectangles for two nuisance parameters, and supremum *P*-values over possible

**Table 2. Newly acquired mutations detected in serial transplantation.**

| Patient | Chromosome | Position | Gene | Altered allele frequency | | | |
|---------|-----------|----------|------|---------|-----|-----|-----|
| | | | | *In vitro* | X1° | X2° | X3° |
| P1 | 1 | 231061360 | *TTC13* | 0.00 | 0.00 | 0.05 | 0.02 |
| P1 | 14 | 20444084 | *OR4K15* | 0.00 | 0.00 | 0.06 | 0.03 |
| P1 | 7 | 149495370 | *SSPO* | 0.00 | 0.00 | 0.11 | 0.02 |
| P1 | 19 | 519401 | *TPGS1* | 0.00 | 0.00 | 0.14 | 0.27 |
| P2 | 10 | 98742139 | *C10orf12* | 0.00 | 0.00 | 0.14 | 0.17 |

X1: first xenograft generation; X2: second xenograft generation; X3: third xenograft generation.

constellations of unobservable count data (Appendix Supplementary Methods). The observed clone sizes were compared to their theoretical distributions to test whether (i) proliferation rates of genetically marked clones within a tumor were identical ($H_{0,P1}$), (ii) proliferation rates in primary or secondary mice were identical to the proliferation rates of the same clones in the next mouse generation ("constant growth rates") ($H_{0,P2}$), and (iii) seeding efficiencies of all clones within a tumor were identical ($H_{0,S}$). An upper 99% confidence bound calculated 96–302 clones to be present in individual primary xenografts, implying a seeding efficiency of 0.0069–1.56% of the initially transplanted marked cells. Seeding efficiencies of clones contributing to secondary tumor formation were shown to be heterogeneous, but at least as high as in primary tumors, and cell amplification within engrafted marked TIC clones profoundly fluctuated during serial transplantation (Appendix Tables S3 and S4; Appendix Supplementary Statistical Results). These data indicate that the small subfraction of PDAC cells that initially engrafted in primary mice had retained a high capability to engraft secondary and tertiary mice, especially when these cells had not contributed to xenograft tumor formation in previous generations. Thus, mathematical modeling indicated substantial changes in the proliferative activity of individual, otherwise homogenous TICs, which predominantly produce non-tumorigenic progeny with very limited or no self-renewal.

**Phenotypic plasticity of PDAC TICs**

We then asked whether TIC function is tightly linked to a stem cell-like phenotype. Ten percent of fetal bovine serum was added to the cultures and the growth factors FGF2, FGF10, and Nodal were withdrawn, manipulations that have been previously shown to induce partial differentiation of PDAC TICs (Hermann *et al*, 2007). Under these conditions, PDAC cells no longer grew in three-dimensional colonies but formed monolayers of elongated and irregularly shaped larger cells (Fig 3A). Markers previously described for TICs (e.g., CD133, ALDH1, SOX2, KLF4) or normal pancreatic progenitors (e.g., SOX9, NOTCH1, HES1) were down-regulated in a patient and culture passage-dependent manner as revealed by comparative gene expression profiling (Gene Expression Omnibus, accession number GSE59118), flow cytometry, and indirect immunofluorescence staining (Fig 3B and C; Appendix Tables S5 and S6). Moreover, in three patient cultures (P2–P4), expression of the epithelial differentiation marker KRT7 was up-regulated 2.3- to 7.7-fold after serum exposure. To test whether the phenotypic induction of a more differentiated phenotype was associated with a loss of tumorigenicity, we compared tumor formation by PDAC cells under stem cell conditions (serum-free medium/cytokines) and differentiating conditions (10% FBS-containing medium/no cytokines). No systematic difference was observed in the ability of PDAC cells derived from either culture condition to form serial tumors or in TIC frequencies measured by limiting dilution (Fig 3D; Table 3). Histology of tumors was identical regardless of culture conditions (Appendix Fig S1B), and clonal TIC dynamics *in vivo* were unaffected (Fig 3E).

Tumorigenicity did not correlate with the expression of pancreatic progenitor or TIC markers at the time of transplantation (Appendix Table S6). Even complete loss of CD133, a marker commonly used to enrich TICs in PDAC and other solid tumors (Singh *et al*, 2003; Hermann *et al*, 2007; O'Brien *et al*, 2007; Ricci-Vitiani *et al*, 2007), was not associated with decreased tumor-forming potential (Fig 3F; Appendix Table S6). To confirm the lack of a tight link between CD133 expression and tumor formation, we stringently sorted untreated or serum-treated PDAC cells according to their CD133 (P2, P3) or combined CD133/CD44 (P1) expression. Strikingly, sorted cell populations were equally able to form tumors after xenotransplantation. Moreover, the tumors grown from highly purified CD133$^-$ and CD133$^+$ cells contained the same proportion of cells expressing CD133 (Appendix Table S6, Appendix Fig S1C) and were serially transplantable. As CD133 was not predictive for

**Figure 3. Phenotypic plasticity of pancreatic TICs.**

A     After the addition of 10% FBS to the culture medium and withdrawal of growth factors, cells grew in monolayers.

B, C   Under these conditions, cells down-regulate SOX2 and CD133 and up-regulate the expression of the duct marker KRT7.

D     Tumor formation was not altered after three passages under differentiation conditions. Cells retained TIC capacity even after 7–8 passages in differentiation culture. Although tumor weights were reduced (*t*-test; *$P$ = 0.00027), tumors formed in 20/24 transplanted mice. Horizontal lines indicate mean.

E     Serial transplantation of patient P1-derived TICs cultured in FBS-containing conditions demonstrated similar clonal succession of transiently active TICs as in serum-free culture conditions.

F     Sorted CD133$^+$ and CD133$^-$ PDAC cells equally efficiently generated tumors.

Data information: Scale bars: (A) 100 μm; (B) 30 μm; and (F) 1 cm. Experiments (A–C) performed in duplicate for patients P1–P4. Representative data from P2 are shown.

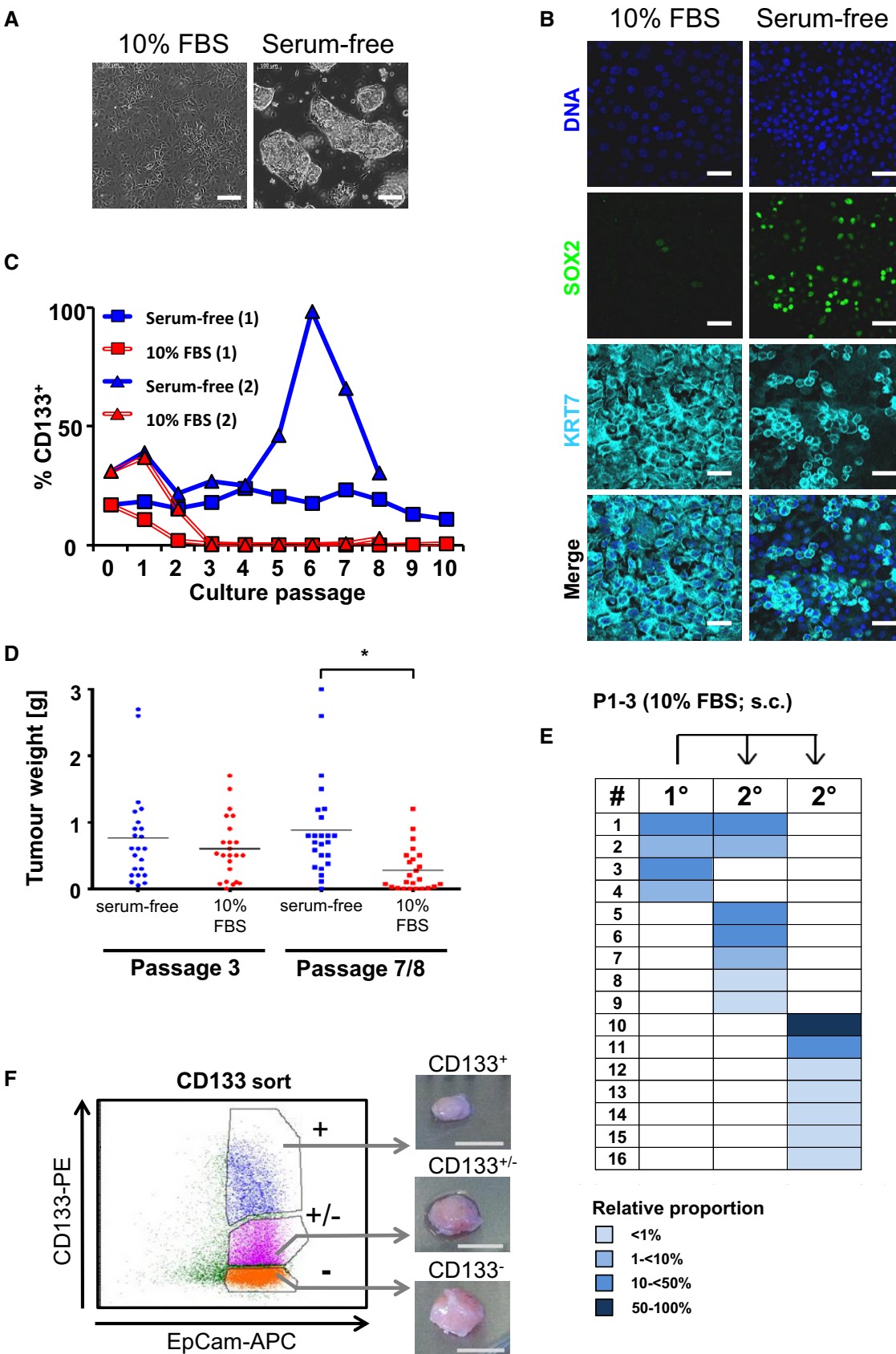

Figure 3.

**Table 3.  TIC frequency assessed in limiting dilution transplantation.**

| Patient experiment a/b | Positive mice/transplanted mice | | | | | | TIC frequency |
|---|---|---|---|---|---|---|---|
| | Treatment | $10^4$ cells | $10^3$ cells | $10^2$ cells | 10 cells | 1 cell | |
| P2-a | Serum-free | 2/4 | 0/4 | 1/4 | 0/4 | 0/4 | 1 in 10,965 |
| | 10% FBS | 4/4 | 1/3 | 0/4 | 0/4 | 0/3 | 1 in 2,341 |
| P2-b | Serum-free | 3/4 | 0/2 | 0/3 | 0/4 | 0/4 | 1 in 8,113 |
| | 10% FBS | 2/4 | 0/2 | 0/4 | 0/4 | 0/3 | 1 in 15,694 |
| P3-a | Serum-free | 4/4 | 4/4 | 2/4 | 1/4 | 0/4 | 1 in 108 |
| | 10% FBS | 4/4 | 3/3 | 2/4 | 0/4 | 0/4 | 1 in 164 |
| P3-b | Serum-free | 4/4 | 4/4 | 3/4 | 2/4 | 0/4 | 1 in 47 |
| | 10% FBS | 4/4 | 4/4 | 4/4 | 1/4 | 0/4 | 1 in 28 |

tumor-forming capacity of cultured human PDAC cells, we expanded tumor pieces derived from four patients (P5, P6, P7, and P8) in NSG mice and sorted CD133-positive and CD133-negative PDAC tumor cells from these *in vivo*-expanded tumor pieces which were not cultivated *in vitro* before. Upon subcutaneous transplantation, both fractions readily formed tumors with similar growth kinetics, further underlining that CD133 expression is not stably linked to tumor-forming capacity of human PDAC cells (Appendix Fig S2). These data demonstrate a pronounced phenotypic plasticity of cells with tumor-initiating capacity in human PDAC.

## Discussion

Our study supports a new model for the organization of the proliferative compartment within a solid cancer, that is, PDAC, in which long-term tumor progression is driven by a succession of transiently active TICs generating tumor cells in temporally restricted bursts (Fig 4). These findings are in stark contrast to the clonal dynamics observed in colorectal cancer and acute myeloid leukemia as previously reported by our group and others. In these malignancies, cancer cell generation is ultimately driven by extensively self-renewing tumor stem cells at the top of hierarchically organized stem cell systems (Dieter *et al*, 2011; Kreso *et al*, 2013).

In pancreatic cancer, tumor-initiating cells have been initially described based on tumor-forming capacity of fractionated cell populations (Hermann *et al*, 2007; Li *et al*, 2011). Importantly, the functional capacity of human TICs cannot be analyzed within the natural undisturbed *in vivo* context, that is, within the patient. Instead, experimental analysis of TIC biology in humans by nature requires surgical removal of cancer tissue, dissociation of the patient tumor, and subsequent functional readouts in adequate *in vivo* and *in vitro* surrogate models. Still, by adapting functional assays originally developed for normal adult stem cells, key properties of TICs have been successfully investigated in such model systems (Dalerba *et al*, 2007; Hermann *et al*, 2007; Li *et al*, 2007; O'Brien *et al*, 2007; Ricci-Vitiani *et al*, 2007). However, bulk transplantation experiments without clonal markers can be largely misleading in this respect as they do not provide the single clone resolution required for unequivocal detection of self-renewal. Furthermore, sorting strategies based on cell surface marker expression would exclude

relevant proportions of tumor cells from further analyses. Clonal marking by lentiviral transduction overcomes these shortcomings and has therefore been widely adopted to monitor the clonal dynamics in normal and more recently malignant regenerative cell compartments. By using this approach, we here describe the clonal dynamics of individually marked tumorigenic PDAC cells in serial transplantation.

Initially, the interpretation of early clonal marking studies was challenged by the limited sensitivity of the methodology for clonal marker detection that failed to detect small clones and therefore underestimated stem cell self-renewal (Lemischka *et al*, 1986; Jordan & Lemischka, 1990; Fraser *et al*, 1992; Lemischka, 1992; Guenechea *et al*, 2001). The problem of limited sensitivity of integration site detection was solved by PCR-based integration site amplification technologies: The highly sensitive LAM-PCR used in our study enables the detection of single copies of vector integrations in 1 μg of DNA (Schmidt *et al*, 2001, 2007). This method was extensively validated (Hacein-Bey-Abina *et al*, 2003; Schmidt *et al*, 2003; Ott *et al*, 2006; Cartier *et al*, 2009; Gabriel *et al*, 2009, 2011; Boztug *et al*, 2010; Paruzynski *et al*, 2010; Stein *et al*, 2010; Aiuti *et al*, 2013; Braun *et al*, 2014) and is now considered the gold standard in gene therapy applications (Schepers *et al*, 2012). Using the same experimental approach as in our study reported here, we have recently characterized the clonal kinetics of TICs in colon cancer (Dieter *et al*, 2011), which were found to be fundamentally different from PDAC. In colon cancer the tumor-initiating cell compartment is functionally heterogeneous but long-term progression during serial xenotransplantation experiments is predominantly driven by constantly proliferating and self-renewing long-term TICs (Dieter *et al*, 2011). The clonal architecture in CRC thereby resembled acute myeloid leukemias that have been analyzed by a similar but less sensitive type of approach (Hope *et al*, 2004). More recent studies in transgenic mouse models of squamous cell skin tumors and intestinal adenomas showed that also in unperturbed tumors only a minority of long-term proliferating tumor cells drive tumor progression thereby demonstrating that the hierarchical organization of tumors is not only detectable when transplantation experiments are done to define TIC activity (Hope *et al*, 2004; Driessens *et al*, 2012; Schepers *et al*, 2012).

In highly malignant cancers such as melanomas or *Eu-myc* transgenic mouse lymphoma, many up to almost every cell within a tumor can regenerate tumors after transplantation and the

## Classical model: Self-renewing LT-TIC compartment

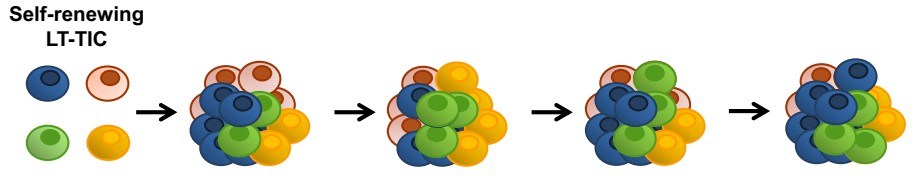

**Self-renewing LT-TIC**

Long-term tumor growth

## Pancreatic cancer: Recruitment of transient TIC activity

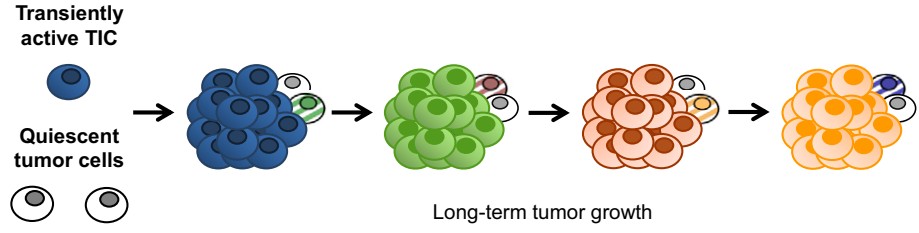

**Transiently active TIC**

**Quiescent tumor cells**

Long-term tumor growth

**Figure 4.  Two alternative models of clonal TIC dynamics underlying long-term growth of epithelial cancers in serial transplantation.**
The classical cancer stem cell model defines extensive self-renewal of long-term TICs (LT-TICs) as driving force within a hierarchically organized TIC compartment. The model of long-term tumor growth in PDAC fundamentally differs from the classical stem cell model in that a succession of TIC clones drives tumor progression in serial xenotransplantation. Individual TICs only transiently contribute to tumor formation and produce mainly non-tumorigenic progeny with very little or no self-renewal. During long-term tumor progression, inactive TIC clones are recruited to tumor formation and outcompete progeny of formerly active TIC clones.

frequency of TICs varies depending on the severity of immunodeficiency of the recipient mouse used (Kelly *et al*, 2007; Quintana *et al*, 2008). In our study, the most immunodeficient recipient mouse strain available to date was used, allowing for efficient engraftment of (tumor) stem cell populations from different organs and tumor entities (Shultz *et al*, 2014). Despite the use of this highly permissive mouse strain, only one out of $10^3$–$10^5$ PDAC cells actively contributed to tumor formation. Strikingly, these clones were not constantly active but lost their tumor-initiating capacity after bursts of cell production. Of note, at each step of serial transplantation, single-cell suspensions have been generated and injected, a strategy that ensures equal distribution of cell clones within the transplant. Nevertheless, the majority of cell clones that contributed strongly to tumor formation in one generation were not detected in subsequent generations. In fact, within a given tumor generation, more than 90% of all marked cells descended from a set of dominant clones and thereby represented the vast majority of all cells which are transplanted into subsequent mice. Although these cells outnumbered all others by far at any given location, they did not contribute to tumor formation in subsequent xenografts in serial transplantation to a measurable extent by our highly sensitive LAM-PCR. This cannot be explained by a stochastic contribution of individual cancer cells but indicates that clonal output was transient and did not depend on intra-tumor localization. This is further supported by our finding that clonal dynamics were similar in serial cultures. In line with this, Ki67 staining of xenografts clearly showed that proliferating cancer cells were equally distributed throughout the tumors.

Mathematical modeling of clone kinetics revealed profound changes in the proliferative activity of individual TICs during serial

transplantation that lead to mainly non-tumorigenic progeny. Taking into account that progeny of large clones in primary tumors were largely overrepresented in the secondary transplants but were nevertheless replaced by formerly inactive clones, these data strongly argue against a purely stochastic process in which at any given time point all cells share the same low likelihood to regenerate tumors after transplantation. In fact, although cancers develop as clonal disease, individual PDAC cells seem to change their functional state by acquiring tumorigenicity during progression that is lost or largely reduced after proliferation. This cannot be explained by a stochastic contribution of all cells; instead, previously highly proliferating cell clones must have substantially slowed down or even lost their proliferative activity. Upon serial transplantation, small clones—which have undergone a limited number of or no cell division(s)—had a selective advantage over highly proliferative clones dominating the precedent tumor mass. Several lines of evidence indicate that the microenvironment within the PDAC tumors did not majorly contribute to the observed clonal dynamics in our experimental setting. First, dynamics of PDAC TICs were independent from the site of transplantation. Second, the *in vitro* kinetics of PDAC cell clones in serially passaged stroma-free cultures were remarkably similar to the *in vivo* kinetics in serially passaged tumors, strongly suggesting that the observed successive transient activation and inactivation of PDAC clones is not dependent on the cellular context within tumors but a cell-intrinsic property of PDAC cells.

Parallel clonal evolution mechanisms have been shown to result in genetic subclones within individual leukemias and solid tumors which can differ in tumor and metastasis formation and may

respond differentially to chemotherapy (Stratton *et al*, 2009; Campbell *et al*, 2010; Meyerson *et al*, 2010; Yachida *et al*, 2010; Navin *et al*, 2011; Notta *et al*, 2011). Moreover, dynamic contributions of genetic subclones have been described in patient-derived xenografts from different tumor entities (Notta *et al*, 2011; Eirew *et al*, 2015). Importantly, exome sequencing of serial pancreatic cancer xenografts within our study did not provide any evidence that such clonal evolution mechanisms significantly contributed to the observed functional clone dynamics. Only a very limited number of acquired additional mutations were detected and there was no evidence for pronounced genetic subclone fluctuations within each xenograft transplantation series indicating that the observed clone kinetics in PDAC reflect changes in the functional state of pancreatic cancer cells that are not driven by genetic mutations.

Our data establish that long-term tumor progression in serial xenotransplantation can be driven by functional plasticity and does not necessarily require stably self-renewing tumor stem cells within a fixed malignant stem cell compartment (Fig 4). Of note, in our study we studied samples derived from a limited number of three PDAC patients but with very similar results. Therefore, we cannot exclude different clonal dynamics in small subgroups of PDAC tumors. A limitation of our study is that continuous tumor growth cannot be assayed in serial xenotransplantation, as tumor explantation and dissociation are required. Therefore, it cannot be ruled out that the procedure of xenotransplantation per se impacts the clonal dynamics observed. However, by using the same methodology we have previously demonstrated a hierarchical organization of the TIC compartment in colorectal cancer (Dieter *et al*, 2011), clearly establishing that the experimental model used is permissive for fundamentally different clonal dynamics of patient samples from different malignant diseases.

In surprising analogy to our data, tissue regeneration in normal pancreas is not driven by a distinct adult stem cell population as in highly proliferative tissues such as the colon or the hematopoietic system. Instead, upon chemical damage, non-proliferating differentiated acinar cells undergo transient de-differentiation into stem-like proliferating progenitor cells, followed by re-differentiation into acinar cells (Fendrich *et al*, 2008; Morris *et al*, 2010). It is tempting to speculate that long-term progression of PDAC is driven by mechanisms operative in the normal post-mitotic exocrine pancreas that allow substitution of damaged cells by previously non-proliferative tumor cells. The unexpected phenotypic and functional plasticity of TICs in PDAC xenografts observed in our study clearly poses challenges for the development of TIC-directed treatment strategies. Phenotype- or mechanism-based targeting of active TIC clones may be counteracted by recruiting TIC activity to substitute eliminated TICs within PDAC tumors. These findings point out the need to develop efficient TIC-directed therapies against functional TIC activity, rather than against a fixed stem cell population, in human PDAC.

## Materials and Methods

### Laboratory animals

Male or female immune-deficient NOD.Cg-Prkdc$^{scid}$Il2rg$^{tm1Wjl}$/SzJ (NSG) mice were purchased from The Jackson Laboratory (Bar Harbor, Maine, USA) and further expanded in the Centralized Laboratory Animal Facilities of the German Cancer Research Center of Heidelberg. Animals were group-housed in standard individually ventilated cages with wood chip embedding (LTE E-001, ABEDD, Vienna, Austria, in Heidelberg), nesting material, *ad libitum* diet (autoclaved mouse/rat housing diet 3437; PROVIMI KLIBA AG, Kaiseraugst, Switzerland), and autoclaved tap water. In accordance with Appendix A of the European Convention for the Protection of Vertebrate Animals used for Experimental and Other Scientific Purposes from March 19, 1986, room temperature and relative humidity were adjusted to $22.0 \pm 2.0°C$ and $55.0 \pm 10.0\%$, respectively. All animals were housed under strict specific pathogen-free (SPF) conditions according to the recommendations of the FELASA. The light/dark (L/D) cycle was adjusted to 14 h lights on and 10 h lights off with the beginning of the light and dark period set at 6.00 am and 8 pm, respectively. All animal experimentation performed in this study was conducted according to the national guidelines and was reviewed and confirmed by an institutional review board/ethics committee headed by the responsible animal welfare officer. The animal experiments were finally approved by the responsible national authority, which is the Regional Authority of Karlsruhe (Germany; approval numbers G-76/12 and G233-15). For all experiments described, a total of 410 eight- to fourteen-week-old male or female NSG mice, weighing 21–30 g, were used.

### *In vivo* expansion of primary human pancreatic tumors

Fresh patient-derived tumor material was received from the Department of Surgery of the Heidelberg University Hospital. Between $5 \times 10^4$ and $2 \times 10^6$ purified cells in 30–50 μl culture medium, mixed 1:1 with Matrigel (BD Biosciences) or whole-tumor pieces of 1–5 mm size, were transplanted subcutaneously or under the kidney capsule of 8- to 14-week-old male or female immune-deficient NOD.Cg-Prkdc$^{scid}$Il2rg$^{tm1Wjl}$/SzJ (NSG) mice. For xenotransplantations, mice were anesthetized on a 37°C head pad with 1.75% isoflurane in the breathing air. As pain killer, 4 ng carprofen was applied per gram body weight. Mice were checked daily and sacrificed by cervical dislocation 3–26 weeks after transplantation. Tumor tissue was removed and further transplanted as whole-tumor pieces subcutaneously or under the kidney capsule of NSG mice for further expansion.

### Purification and culture of primary human pancreatic cancer cells

Tumor tissue was minced into small pieces (< 2 mm) and digested with 2 mg/ml collagenase IV in Medium 199 (Invitrogen) containing 3 mM CaCl$_2$ for 2.5 h. Subsequently, purified tumor cells were washed with D-PBS (Invitrogen) and filtered once though a 100-μm and twice through a 40-μm cell strainer (BD Biosciences) to receive a single-cell suspension. For serum-free culture conditions, DMEM/F12 medium was used supplied with 6 mg/ml glucose, 2 mM L-glutamine, 1% penicillin/streptomycin, B27 supplement (Invitrogen), 5 mM HEPES buffer, 6 μg/ml heparin (Sigma-Aldrich), 10 ng/ml FGF2, 20 ng/ml FGF10, and 20 ng/ml Nodal (R&D Systems). To establish spheroid cultures, purified single-cell suspensions were taken into serum-free medium in ultra-low attachment plates (Corning). For adherent cultures, undigested whole-tumor pieces were

plated into normal cell culture flasks (Thermo scientific) to allow attachment of tumor cells. For phenotypic differentiation, adherent cells were detached by Accutase (PAA) and replated in RPMI 1640 medium supplemented with 2 mM L-glutamine, 1% penicillin/strep-tomycin (Invitrogen), and 10% fetal bovine serum (FBS) (PAA). Cultures were split 1:1 to 1:5 by Accutase (PAA)-mediated cell detachment. Cytokines were added every 3–4 days to serum-free cultures. Established primary TIC cultures were regularly tested mycoplasma-free.

## Serial transplantation and *in vitro* passaging of primary pancreatic tumor cells

Between 10 and $1 \times 10^7$ transduced or non-transduced cells derived from four individual patients (P1–P4) were collected in 30–50 μl culture medium, mixed 1:1 with Matrigel (BD Biosciences), and transplanted subcutaneously, under the kidney capsule or into the pancreas of immune-deficient NSG mice in non-blinded, unrandom-ized single-arm explorative studies. Mice were monitored daily and sacrificed by cervical dislocation 3 weeks to 12 months after trans-plantation to harvest the grown tumor tissue. For serial transplanta-tion, tumor cells were purified as described above and 33–50% of all cells (total of $5 \times 10^3$ to $1 \times 10^7$ cells) were transplanted into secondary or tertiary recipient mice. For analyzing clonal kinetics *in vitro*, explanted tumor cells or freshly transduced adherent cultures were cultured until they reached confluence and then split 1:10 for up to eight culture passages. The remaining cells were frozen at −80°C and subjected to integration site analysis.

For limiting dilution transplantation, decreasing cell numbers passaged for three passages in FBS-containing or serum-free culture conditions were transplanted subcutaneously in NSG mice. Mice were examined regularly for tumor formation and sacrificed when tumors reached the maximum tolerable size or when experiments were ended. In case of small tumors, histology was confirmed. Frequencies of tumor-initiating cells and differences between the treatment groups were calculated using Poisson statistics and the method of maximum likelihood (ELDA web tool; Hu & Smyth, 2009). Mice perished without tumor formation before experiment was ended were excluded from analysis.

## Immunofluorescence staining

Adherent cultures were grown on coverslips (Geyer) and fixed with 4% paraformaldehyde (Roth) in PBS (Invitrogen). Cells were blocked in PBS + 0.1% BSA (Sigma-Aldrich) and permeabilized using 0.1% Triton X-100 (AppliChem) and 0.1% sodium citrate (Sigma-Aldrich) in PBS. DNA was stained by Hoechst (Invitrogen), and the actin cytoskeleton was visualized by phalloidin-PF647 (Promokine). For marker staining, cells were incubated with the following primary antibodies (dilution 1:100, if not stated other-wise) for 1–16 h in a wet chamber: rabbit anti-human KRT7 (1:200) (product code ab53123; Abcam); rabbit anti-OCT4 (product code ab19857; Abcam); mouse anti-human KLF4 (clone 56CT5.1.6; Abgent); mouse anti-E-cadherin (1:50) (clone 36/E-Cadherin; BD Biosciences); rat anti-mouse thy1.2-PE (1:50) (clone 53-2.1; BD Biosciences); mouse anti-human KRT7 (clone OV-TL 12/30; Dako); goat anti-human PTF1a (catalog number AF6119; R&D Systems); goat anti-human SOX2 (catalog number AF2018; R&D Systems);

mouse anti-vimentin (clone V9; Santa Cruz); rabbit anti-α-amylase (catalog number A8273; Sigma-Aldrich); rabbit anti-Zeb1 (catalog number HPA027524; Sigma-Aldrich); and rabbit anti-β-catenin (cat-alog number C2206; Sigma-Aldrich). Subsequently, cells were incu-bated with the following secondary antibodies for 1 h, dilution 1:200: donkey anti-mouse IgG-DyLight 649 (code number 715-495-150 Jackson IR); donkey anti-rabbit IgG-DyLight 549 (code number 711-505-152; Jackson IR); donkey anti-goat IgG-DyLight 488 (code number 705-485-147; Jackson IR); donkey anti-rat IgG-DyLight 649 (code number 712-495-150; Jackson IR); goat anti-mouse IgG-PF555 (catalog number PC-PK-PF555-AK-M1; Promokine); and goat anti-rabbit IgG-PF488 (catalog number PK-PF488P-AK-R1; Promokine).

## Histopathology and immunohistochemistry

For histological and immunohistochemical analysis, tumor tissue or tumor spheres were fixed with 10% formalin solution (Sigma-Aldrich) and embedded in paraffin (Merck) or 30% albumin (Serva). Ten-micrometer sections were stained with hematoxylin and eosin. The histological appearance of the tumor tissue was analyzed by a senior pathologist specialized in pancreas-specific pathology. For immunohistochemistry, sections were stained with primary antibodies rabbit anti-KRT7 (dilution 1:200, product code ab53123; Abcam), mouse anti-vimentin (dilution 1:100, clone V9; Santa Cruz), and rabbit anti-Ki67 (dilution 1:500, product code ab15580; Abcam) and the secondary antibodies goat anti-mouse IgG-PF555 (dilution 1:200, catalog number PC-PK-PF555-AK-M1; Promokine) and goat anti-rabbit IgG-PF488 (dilution 1:200, catalog number PK-PF488P-AK-R1; Promokine). DNA in the nuclei was visualized with Hoechst (Invitrogen). Note that in the presence of albumin, Hoechst staining can generate increased background fluo-rescence. For phenotyping and fluorescence-activated cell sorting (FACS), cells were resuspended in Hanks' balanced salt solution (HBSS) (Sigma-Aldrich) supplemented with 2% fetal bovine serum (FBS) (Pan Biotechnology). Primary pancreatic tumor cells were stained for 30 min on ice with antibodies against human CD44 (clone G44-26, dilution 1:10, or 27–35, dilution 1:200; BD Bios-ciences), human CD24 (clone ML5, dilution 1:20; BD Biosciences), human EpCam (clone EBA-1, dilution 1:20; BD Biosciences), human CD133 (clone AC133 and 293C3, dilution 1:10; Miltenyi), human thy1 (clone 5E10, dilution 1:200; BD Biosciences), and murine thy1.2 (clone 53-2.1, dilution 1:50; BD Biosciences). To enhance the efficiency of CD133 detection, AC133 and 293C3 antibodies were used in combination. Dead cells were excluded by washing with 0.2 μg/ml propidium iodide in HBSS + 2% FBS. Surface marker expression was analyzed by flow cytometry (LSRII or AriaII; Becton Dickinson) using FACSDIVA software (Becton Dickinson).

## Lentiviral marking and LAM-PCR integration site analysis

For lentiviral marking with minimal insertional activation of neigh-boring genes, third-generation self-inactivating lentiviral vectors were used, which were pseudotyped with the glycoprotein of the vesicular stomatitis virus (VSV-G). The vectors encoded for EGFP under the control of the human PGK promoter. Vector supernatant was produced as described previously (Dull *et al*, 1998) using the transfection reagent polyethylenimine (Sigma-Aldrich). Vector parti-cles were concentrated by ultracentrifugation. For transduction,

cultured cells were digested to a single-cell suspension using Accutase (PAA). Up to $2 \times 10^6$ tumor cells were transduced overnight under regular cell culture conditions with efficiencies ranging from 35.5% to 97.5% (P1–P3) and transplanted within 24 h. Primary cells derived from P4 could not be efficiently transduced and therefore were excluded from subsequent analyses. After tumor formation, the lentivirally marked tissue was purified as described above. Between 1/3 and 1/2 of the purified tumor cell suspension was subjected to DNA isolation using the DNeasy Blood & Tissue Kit (Qiagen). 500 ng to 1 μg of resulting DNA was used for integration site analysis by 3′LTR-LAM-PCR, and subsequent 454 pyrosequencing (GS Junior; Roche Diagnostics) as well as analysis of sequenced integration sites was done as described previously (Schmidt *et al*, 2007; Dieter *et al*, 2011). For sequencing, specific barcoded primers were added to both ends of the LAM-PCR. LAM-PCR was performed twice using either TSP509I or MSEI to enlarge coverage of genomic integrations. The sequenced integration sites were valid if the lentiviral LTR was fully present at the vector-genome junction and the genomic DNA flanking the vector integration had a unique sequence match of ≥ 95% when aligned to the human genome. A sequencing validity cutoff was set based on sequencing of clones harboring known integration sites; the cutoff was defined at a minimal level below 0.06% of total sequence counts. Collisions below the cutoff were subsequently removed from the analysis.

## Gene expression profiling

Primary pancreatic tumor cell cultures were grown in parallel for up to 10 passages under serum-free or 10% FBS-containing conditions as described above. At passages 3 and 8, total RNA of the cells was isolated using the RNeasy Mini Kit (Qiagen). The RNA was analyzed by comparative gene expression profiling using HumanHT-12 v4 Expression BeadChip technology (Illumina), and data were analyzed employing Chipster software (Kallio *et al*, 2011).

## Whole-exome sequencing

DNA was isolated from lentiviral marked xenografts, primary adherent TIC cultures, and corresponding blood samples or normal pancreatic tissue using DNeasy Blood & Tissue Kit or QIAamp DNA Mini Kit (Qiagen). DNA concentrations were measured by Qubit™ dsDNA BR Assay Kit 500 (Life Technologies). Authentication test was done using the Investigator ESSplex Plus Kit (Qiagen), and quality was verified using the Agilent 2200 TapeStation. Library preparation was done on the Agilent NGS Workstation (version F.0) according to the manufacturer's protocol using the SureSelectXT Automation Reagent Kit (Agilent) and Human All Exon V5 + UTRs (Agilent). Paired-end sequencing ($2 \times 100$ bp) was done on a HiSeq2000 applying TruSeq PE Cluster Kit v3 and TruSeq SBS Kit v3 (Illumina). Paired-end DNA sequencing reads were aligned similar to the methods described previously (Jones *et al*, 2013). Briefly, reads were mapped to a concatenated human and murine genome assembly to reduce false-positive mutations introduced by murine contamination in the xenograft samples [1000 Genomes phase 2 reference assembly (hs37d5) (ftp://ftp.1000genomes.ebi.ac.uk/vol1/ftp/technical/reference/phase2_reference_assembly_sequence/) and the UCSC mm10 genome assembly (http://hgdownload.cse.ucsc.

edu/goldenPath/mm10/bigZips/chromFa.tar.gz)]. Reads were aligned using BWA aln (version 0.6.2-r126-tpx) with a Q20 Phred score quality threshold for read trimming down to 35 bp and using 8 threads (-q 20-t 8). Subsequent alignments were merged using samtools merge (version 0.1.19-44428cd) (40) and PCR duplicates were marked using Picards tools (version 1.61-1094) (http://picard.sourceforge.net, 2014). Single nucleotide variants (SNVs) and small insertions/deletions (indels) were identified using an in-house analysis pipeline based on samtools mpileup, and bcftools (Naldini, 2009). SNVs were identified using samtools mpileup in the tumor DNA sample by counting anomalous read pairs, disabling BAQ computation, using reads with a minimum mapping quality of Q30, and using bases with a minimum base quality of Q23; these were filtered for non-novel mutations by comparison with the appropriate preceding sample. All novel non-silent SNVs were manually inspected using IGV (Thorvaldsdottir *et al*, 2013) to verify that the mutations were not artefactual of local indels, structural variations, misaligned indels, and possible murine contamination. The possible murine contaminants were identified as clusters of novel SNVs only observed in the xenograft samples. All high-confidence non-silent mutations found to be novel in any stage of progression of the sample were identified. For each of these positions, the bases aligning to each of these positions for each sample were extracted using samtools mpileup, filtering for reads using anomalous pairs, with disabled BAQ computation, with minimum mapping quality of 30, and minimum base quality of 23 (-A -B -R -q 30 -Q 23). These mutational positions were filtered for a minimum read depth of 20× across each sample (apart from CDKN2A) and for mutations that occurred in dbSNP (v138) that were not implicated in disease. The reference and alternate allele frequencies were calculated and used to evaluate the possibility of clonal propagation. VarScan2 was used to identify somatic CNAs in the exome samples (with the preceding sample used as the reference control) using the recommended workflow (Koboldt *et al*, 2012).

## Data availability statement

Gene expression profiling data have been deposited in NCBI's Gene Expression Omnibus and are accessible through GEO Series accession number GSE59118 (http://www.ncbi.nlm.nih.gov/geo/query/acc.cgi?acc = GSE59118). Whole-exome genome data have been deposited at the European Genome-phenome Archive (EGA, http://www.ebi.ac.uk/ega/) which is hosted at the EBI, accession number EGAS00001000882.

## Ethics statement

All experiments with human material were done in accordance with the guidelines of the declaration of Helsinki. Informed consent was received from participants prior to inclusion in the study (Ethikkommission Medizinische Fakultät Heidelberg, approval number 323/2004). Mice were kept in a specific pathogen-free animal facility according to German laws and permission of the institutional ethic committee (Regierungspräsidium Karlsruhe, approval number G-76/12). The ARRIVE guidelines were consulted.

**Expanded View** for this article is available online.

## The paper explained

### Problem

Pancreatic ductal adenocarcinoma (PDAC) is one of the deadliest human cancers due to frequent local recurrence and metastases. Experimental models point to a hierarchical cellular organization of PDAC with tumor-initiating cells (TICs; also referred to as cancer stem cells) that drive tumor progression and metastasis formation. Although TIC self-renewal has been postulated to be essential in progression and metastasis formation of human PDAC, clonal dynamics of TIC within PDAC tumors are yet unknown.

### Results

By tracking the clonal contribution of PDAC TICs to tumor formation *in vivo*, we found that individual PDAC xenograft tumors stem from distinct sets of TICs with very little overlap between subsequent generations of tumors. The recruitment of inactive TIC clones to tumor formation by serial transplantation indicates an unexpected functional and phenotypic plasticity of pancreatic TICs *in vivo*. These data indicate that long-term progression of PDAC is driven by a succession of transiently active TICs producing tumor cells in temporally restricted bursts.

### Impact

The observed clonal succession of TIC activity in PDAC xenografts is in stark contrast to the continuous activity of limited numbers of self-renewing TICs within a fixed cellular hierarchy observed in other epithelial cancers. Our data show that the regulation of TIC activity, rather than a fixed TIC population, needs to be targeted for efficient treatment strategies against PDAC.

## Acknowledgements

The authors thankfully acknowledge Sylvia Fessler, Sabrina Hettinger, Galina Dornhof, Stefanie Wenzel, Tim Kindinger, and Annika Mengering for technical assistance and Prof. Annette Kopp-Schneider for assistance in the mathematical modeling of clone growth. The authors thank the DKFZ Central Animal Laboratory for animal care, the DKFZ Genomics and Proteomics Core Facility for sequencing, the DKFZ Light Microscopy Facility for imaging, the NCT Tissue Bank for providing tissue in accordance with the formalities of the NCT Tissue Bank as permitted by the University Ethics Review Board, and the DKFZ-Heidelberg Center for Personalized Oncology (DKFZ-HIPO) through HIPO-00 for whole-exome sequencing. This work was supported by the Deutsche Forschungsgemeinschaft (SFB873), the NCT3.0 Precision Oncology Program (NCT3.0_2015.4 TransOnco), the EU framework programme Horizon2020 (TRANSCAN-2 ERA-NET) and the German Cancer Aid (Colon-Resist-Net).

## Author contributions

CRB, FO, and KRE designed the project, performed experiments, collected and analyzed the data, and wrote the manuscript; TDD, CMH, and SMD performed experiments and analyzed the data; UA performed statistical analysis; MK, JWer, and JWei provided primary human tumor tissue and clinical information; FB and WW prepared histology samples and provided pathology analysis; NI and BB performed bioinformatical analyses of exome sequencing data; FH, MS, and CvK analyzed the data; CS and SF analyzed the data and helped writing the manuscript; and HG designed the project, analyzed the data, and wrote and finally approved the manuscript.

## Conflict of interest

The authors declare that they have no conflict of interest.

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
