## [Review Process File · EMBO Molecular Medicine]

Succession of Transiently Active Tumor-Initiating Cell Clones in Human Pancreatic Cancer Xenografts

Claudia R. Ball, Felix Oppel, K. Roland Ehrenberg, Taronish D. Dubash, Sebastian M. Dieter, Christopher M. Hoffmann, Ulrich Abel, Friederike Herbst, Moritz Koch, Jens Werner, Frank Bergmann, Naveed Ishaque, Manfred Schmidt, Christof von Kalle, Claudia Scholl, Stefan Fröhling, Benedikt Brors, Wilko Weichert, Jürgen Weitz, Hanno Glimm

Corresponding author: Hanno Glimm, National Center for Tumor Diseases NCT

Review timeline:	Submission date:	25 November 2016
	Editorial Decision:	21 December 2016
	Revision received:	09 March 2017
	Editorial Decision:	29 March 2017
	Revision received:	11 April 2017
	Accepted:	21 April 2017

Transaction Report:

Editor: Roberto Buccione

1st Editorial Decision

21 December 2016

Thank you for the submission of your manuscript to EMBO Molecular Medicine.

I apologise for the delay in providing you with a decision. We experienced significant difficulties in securing willing and appropriate reviewers and then obtaining their evaluations in a timely fashion. Also, I have not been able to obtain a third evaluation on this manuscript. Hence I have decided to proceed based on the two available evaluations to avoid further delays.

As you will see, while reviewer 2 appears more positive, reviewer 1 is rather more reserved. Specifically, reviewer 1's main concern is the that transplantation procedure and a number of parameters related to this, heavily affect the outcome and consequent interpretation of the results and does not feel that the current dataset actually informs us on hierarchical and clonal architecture. Reviewer 2, while less reserved, does ask two poignant questions, complementary and partially overlapping with reviewer 1's concerns: the relationship between clonal dynamics and metastatic growth and the fact that the limited number of tumors considered might not allow for generalization.

After further cross-commenting with the reviewers it was agreed that the study is potentially very interesting but it is currently lacking in conclusiveness and sufficient experimental support for the main claims.

In conclusion, I am prepared to consider a substantially revised submission, with the understanding

that the Reviewers' concerns must be addressed, especially to verify the relevance of your findings by comparing clonal representation in primary tumors and metastasis, since metastasis may be a physiological process that somewhat recapitulates an experimental tumor initiation setting. As for the other limitations of the study, these should be at a minimum, properly acknowledged and appropriately discussed.

Please note that it is EMBO Molecular Medicine policy to allow a single round of revision only and that, therefore, acceptance or rejection of the manuscript will depend on the completeness of your responses included in the next, final version of the manuscript.

As you know, EMBO Molecular Medicine has a "scooping protection" policy, whereby similar findings that are published by others during review or revision are not a criterion for rejection. However, I do ask you to get in touch with us after three months if you have not completed your revision, to update us on the status. Please also contact us as soon as possible if similar work is published elsewhere.

Please note that EMBO Molecular Medicine now requires a complete author checklist (<http://embomolmed.embopress.org/authorguide#editorial3>) to be submitted with all revised manuscripts. Provision of the author checklist is mandatory at revision stage; The checklist is designed to enhance and standardize reporting of key information in research papers and to support reanalysis and repetition of experiments by the community. The list covers key information for figure panels and captions and focuses on statistics, the reporting of reagents, animal models and human subject-derived data, as well as guidance to optimise data accessibility.

We now mandate that all corresponding authors list an ORCID digital identifier. You may acquire one through our web platform upon submission and the procedure takes <90 seconds to complete. We also encourage co-authors to supply an ORCID identifier, which will be linked to their name for unambiguous name identification.

I look forward to seeing a revised form of your manuscript as soon as possible.

***** Reviewer's comments *****

Referee #1 (Comments on Novelty/Model System):

The model system is suitable to study tumor initiation in immune compromised mice but has no relevance for established cancer tissue as is claimed.

Referee #1 (Remarks):

With interest I read the manuscript "Succession of transiently active tumor-initiating cell clones in human pancreatic cancer" by Claudia R. Ball et al.

The authors clonally marked primary human pancreatic ductal adenocarcinoma (PDAC) cells by lentiviral transduction and serially transplanted these xenografts. With this model the authors aim to monitor the clonal dynamics and self-renewal activity of clones during tumor formation in vivo. Rather surprisingly, by analyzing the different integration sites of the vector, using LAM-PCR, they found that almost no overlap in integration sites between the cells of the primary, secondary and tertiary xenografts was present. The authors conclude this indicates distinct transiently active populations of TICs are present, generating xenograft tumors after transplantation in PDAC xenografts.

The manuscript is well written and the experiments well described and have been executed to the highest standards. However, I have some serious doubts about the interpretation of the findings.

Major comments/recommendations

- The most important criticism involves the transplantation procedure itself. It is concluded that the continuous appearance of 'new' clones is due to differential activation of various TIC populations, and potential exhaustion of previous highly proliferative clones.

Firstly, the cells that are analyzed for their integration side are by nature of the procedure not the

cells that are serially transplanted. Hence, there is an important selection taking place here. Secondly, and more importantly, the stochasticity of the seeding determines the eventual results of the clonal outgrowth. Apparently only a very limited number of cells after injection substantially contributes to each xenograft, and a large proportion of cells simply sit there because they are on the inside of the tumour(?) and are quickly outgrown by cells on the outside of the tumour. This has nothing to do with stem cell potential.

- In relation to the point above, even if based on the current data conclusions can be drawn on stem cell function in cancer this only involved the initiation phase of xenografts not the dynamics of stem cells in established cancer tissue. This is an important drawback.

- It would be interesting to see if a difference in proliferation rate can be shown in the xenograft tumors and if there are specific regions where there is more or less proliferation. Now solely based on mathematical analysis of the data, this proliferation difference was explained. However, this was not clearly evident from the experiments; could the researchers understate their findings with an experiment (Ki67 staining)? This could possibly strengthen their findings.

- Did the authors find metastases of the renal capsule and orthotopic PDAC xenograft transplantations? According to their previous paper (Dieter et al, 2011), where they examined the TIC in human colon cancer, they showed that metastasis formation was almost exclusively driven by self-renewing long-term-TICs. It would be interesting to see which population of TIC is driving metastases in PDAC as well.

- I think the plural TICs should be used at many instances instead of TIC.

Referee #2 (Remarks):

"Succession of Transiently Active Tumor-Initiating Cell Clones in Human Pancreatic Cancer"

Previous studies had identified populations of tumor initiating cells in pancreas tumors with the capacity to propagate the disease in mice. It was thus proposed that pancreas cancer follows a hierarchical organization similar to that present in other cancer types. These assays, however, were largely biased by the choice of surface markers used to identify and isolate tumor cell populations. In addition, tumor initiation assays only provide information about the behavior of isolated tumor cells. In this manuscript, Glimm and colleagues used lentiviral marking of primary pancreatic cancer cultures to study tumor growth. In many aspects, this approach is superior to transplantation assays as it enables an unbiased assessment of clone dynamics on whole tumors. Authors had previously applied a similar approach to study Colorectal Cancer (CRC) where they reached the conclusion that CRCs are maintained by a hierarchy of cells that includes cancer stem cells at the apex. Therefore, clonal dynamics assessed through lentiviral marking validated the model that had been previously elaborated from tumor cell transplantation assays in CRC. Using an equivalent approach, authors now reach the surprising conclusion that Pancreatic cancer is not organized according to a hierarchy but rather that long-term growth in this type of tumors is maintained by successive recruitment of inactive clones that are activated in restricted periods. Fittingly, authors also demonstrate that frequencies of TICs in ex vivo cultures do not correlate with expression of stem or differentiation markers. Altogether, these are unexpected observations with profound implications to understand pancreatic cancer and improve therapeutic treatments. In my opinion, data are convincing and of high quality. I only have minor criticism/suggestions:

- Does metastatic growth follow clonal dynamics similar to the observed for primary xenografts? This is an important issue as pancreatic cancer is highly metastatic and therapies are not effective to treat metastasis.

- Authors explore the behavior of a small number of primary tumors and therefore it is possible that author's conclusions only hold for a subset of pancreatic cancers whereas other types may follow a hierarchical organization. Authors should remain open to this possibility in the discussion.

Referee #1 (Comments on Novelty/Model System):

The model system is suitable to study tumor initiation in immune compromised mice but has no relevance for established cancer tissue as is claimed.

TICs have been described to drive development and progression of a variety of human cancers, including pancreatic cancer. We agree with the Referee that the functional capacity of human TIC cannot be analyzed within the natural undisturbed *in vivo* situation, i.e. within the patient. Instead, experimental analysis of TIC biology in humans by nature requires surgical removal of cancer tissue, dissociation of the patient tumor and subsequent functional readouts in adequate *in vivo* and *in vitro* surrogate models. Still, by adapting functional assays originally developed for normal adult stem cells, key properties of TIC have been successfully investigated (O'Brien et al Nature 2007, Ricci-Vitiani et al Nature 2007; Dalerba et al, PNAS 2007; Li et al, Cancer Res 2007; Hermann et al. Cell Stem Cell 2007). In this context, serial transplantation of human cancer cells into immunodeficient mice has been widely used to quantify self-renewal of TIC and tumor long-term progression. We adapted these well-established approaches within our study and unexpectedly demonstrate that PDAC progression in this model unlike colon cancer does not require stably self-renewing tumor stem cells within a fixed malignant stem cell compartment. We believe that these findings are highly relevant for the development of therapeutic approaches targeting TIC activity in pancreatic cancer. We added a paragraph to discuss the limitations of the experimental model used.

Referee #1 (Remarks):

With interest I read the manuscript "Succession of transiently active tumor-initiating cell clones in human pancreatic cancer" by Claudia R. Ball et al.

The authors clonally marked primary human pancreatic ductal adenocarcinoma (PDAC) cells by lentiviral transduction and serially transplanted these xenografts. With this model the authors aim to monitor the clonal dynamics and self-renewal activity of clones during tumor formation *in vivo*. Rather surprisingly, by analyzing the different integration sites of the vector, using LAM-PCR, they found that almost no overlap in integration sites between the cells of the primary, secondary and tertiary xenografts was present. The authors conclude this indicates distinct transiently active populations of TICs are present, generating xenograft tumors after transplantation in PDAC xenografts.

The manuscript is well written and the experiments well described and have been executed to the highest standards.

We thank the Reviewer for his appreciation of our experimental work.

However, I have some serious doubts about the interpretation of the findings.

Major comments/recommendations

1. The most important criticism involves the transplantation procedure itself. It is concluded that the continuous appearance of 'new' clones is due to differential activation of various TIC populations, and potential exhaustion of previous highly proliferative clones.

Firstly, the cells that are analyzed for their integration site are by nature of the procedure not the cells that are serially transplanted. Hence, there is an important selection taking place here. Secondly, and more importantly, the stochasticity of the seeding determines the eventual results of the clonal outgrowth. Apparently only a very limited number of cells after injection substantially contributes to each xenograft, and a large proportion of cells simply sit there because they are on the inside of the tumour(?) and are quickly outgrown by cells on the outside of the tumour. This has nothing to do with stem cell potential.

Testing engraftment and cellular progeny generation in serial transplantation is a long-proven standard approach for assaying stem cell activity within a given cell population. In combination with clonal marking, this strategy has been conclusively shown to enable tracing clonal activity of individual normal and benign stem cells *in vivo*. Of note, using the same experimental approach as described here, we recently characterized the organization of the tumor-initiating cell (TIC) compartment in human colon cancer and came to fundamentally different results. Our highly sensitive technology was able to precisely determine the clonal contribution within a hierarchically organized TIC compartment with self-renewing TIC that give rise to transient tumor-amplifying cells in colon cancer xenografts (Dieter, S.M., et al., Cell Stem Cell, 2011. 9(4): p. 357-65).

In the study presented here, the majority of marked PDAC cell clones that strongly contributed to tumor formation in one generation were not detectable in other serial xenograft generations. Importantly, at each step of serial transplantation, single cell suspensions have been generated and injected, a strategy that ensures equal distribution of cell clones within the transplant. Nevertheless, the majority of cell clones that contributed strongly to tumor formation

in one generation were not detected in subsequent generations. As within a given tumor generation, more than 90% of all marked cells descended from a set of tumor-specific dominant clones and thereby represented the vast majority of all cells which are transplanted into subsequent mice this is a highly surprising and unexpected finding. Although these cells outnumber all others by far at any given location they did not contribute to tumor-formation in subsequent xenografts after serial transplantation to a measurable extent by our highly sensitive LAM PCR. This cannot be explained by a stochastic contribution of individual cancer cells but indicate that clonal output was transient and did not depend on intra-tumor localization. In line with this, Ki67 staining of xenografts that we now added as suggested by the reviewer clearly shows that proliferating cancer cells are equally distributed throughout the tumors. In addition, our *in vitro* data demonstrate similar clonal dynamics in serial cultures that by nature cannot be explained by the localization of cells within tumors.

We adjusted the manuscript to make these points clearer.

2. In relation to the point above, even if based on the current data conclusions can be drawn on stem cell function in cancer this only involved the initiation phase of xenografts not the dynamics of stem cells in established cancer tissue. This is an important drawback.

We agree that serial transplantation of human cells in a xenograft model requires disruption of the established cancer tissue by singularization of cells and subsequent injection into the xenogeneic graft. However, despite these limitations, so far this is the only reliable assay to monitor human (tumor) stem cell activity *in vivo* as dynamics of human stem cells cannot be assayed directly in patients. We would like to point out that serial transplantation of clonally marked cells is well established in the field of stem cell research to determine the self-renewal capacity and long-term growth of initially transplanted cell populations (please see also answer to point 1 raised by the same referee). As stated above, we added a discussion of the limitations of the experimental model used to the revised version of the manuscript.

3. It would be interesting to see if a difference in proliferation rate can be shown in the xenograft tumors and if there are specific regions where there is more or less proliferation. Now solely based on mathematical analysis of the data, this proliferation difference was explained. However, this was not clearly evident from the experiments; could the researchers understate their findings with an experiment (Ki67 staining)? This could possibly strengthen their findings.

We thank the Referee for this valuable suggestion. We now have performed Ki67 staining of mouse PDAC xenografts as requested. These data clearly demonstrate the presence of cells with different proliferative activities, i.e. actively proliferating and inactive cells. Strikingly, proliferating cells are equally distributed throughout the tumor, showing that active proliferation is present within the whole tumor mass and not restricted to specific sites of the tumor, i.e. the outside as suggested in comment 1 of the same Referee. As discussed in our response to point 1 these results strongly support our conclusion that the differences in proliferative activity of clones in serial transplantation cannot simply be explained by their spatial distribution, especially as dominant TIC clones make up the vast majority of cells at any given location within the tumor and thereby also supply the vast majority of cells re-transplanted.

4. Did the authors find metastases of the renal capsule and orthotopic PDAC xenograft transplantations? According to their previous paper (Dieter et al, 2011), where they examined the TIC in human colon cancer, they showed that metastasis formation was almost exclusively driven by self-renewing long-term-TICs. It would be interesting to see which population of TIC is driving metastases in PDAC as well.

We agree that understanding the clonal dynamics of metastasis formation would be highly interesting. However, unlike our colon cancer models, the human pancreatic cancer xenografts do not metastasize spontaneously. Accordingly, clonal analysis can only be done on PDAC xenograft tumors and clonal analysis of metastasis formation is simply not possible. Of note, a potential development of metastasis formation surrogate models, subsequent serial transplantation and molecular analyses if successful would require several additional years of experimentation in rather artificial models. We therefore feel that such analyses if at all feasible are –although highly interesting- beyond the scope of the current manuscript and need to be addressed separately.

5. I think the plural TICs should be used at many instances instead of TIC.

We agree and changed the manuscript accordingly.

Referee #2 (Remarks):

"Succession of Transiently Active Tumor-Initiating Cell Clones in Human Pancreatic Cancer"

Previous studies had identified populations of tumor initiating cells in pancreas tumors with the capacity to propagate the disease in mice. It was thus proposed that pancreas cancer follows a hierarchical organization similar to that present in other cancer types. These assays, however, were largely biased by the choice of surface markers used to identify and isolate tumor cell populations. In addition, tumor initiation assays only provide information about the behavior of isolated tumor cells. In this manuscript, Glimm and colleagues used lentiviral marking of primary pancreatic cancer cultures to study tumor growth. In many aspects, this approach is superior to transplantation assays as it enables an unbiased assessment of clone dynamics on whole tumors. Authors had previously applied a similar approach to study Colorectal Cancer (CRC) where they reached the conclusion that CRCs are maintained by a hierarchy of cells that includes cancer stem cells at the apex. Therefore, clonal dynamics assessed through lentiviral marking validated the model that had been previously elaborated from tumor cell transplantation assays in CRC. Using an equivalent approach, authors now reach the surprising conclusion that Pancreatic cancer is not organized according to a hierarchy but rather that long-term growth in this type of tumors is maintained by successive recruitment of inactive clones that are activated in restricted periods. Fittingly, authors also demonstrate that frequencies of TICs in ex vivo cultures do not correlate with expression of stem or differentiation markers. Altogether, these are unexpected observations with profound implications to understand pancreatic cancer and improve therapeutic treatments.

We thank the Reviewer for his precise synopsis.

In my opinion, data are convincing and of high quality. I only have minor criticism/suggestions:

- Does metastatic growth follow clonal dynamics similar to the observed for primary xenografts? This is an important issue as pancreatic cancer is highly metastatic and therapies are not effective to treat metastasis.

We agree that understanding the clonal dynamics of metastasis formation would be highly interesting. However, unlike our colon cancer models, the human pancreatic cancer xenografts do not metastasize spontaneously. Accordingly, clonal analysis can only be done on PDAC xenograft tumors and clonal analysis of metastasis formation is simply not possible. Of note, a potential development of metastasis formation surrogate models, subsequent serial transplantation and molecular analyses if successful would require several additional years of experimentation in rather artificial models. We therefore feel that such analyses if at all feasible are –although highly interesting- beyond the scope of the current manuscript and need to be addressed separately.

- Authors explore the behavior of a small number of primary tumors and therefore it is possible that author's conclusions only hold for a subset of pancreatic cancers whereas other types may follow a hierarchical organization. Authors should remain open to this possibility in the discussion.

We agree with the Reviewer and adjusted the discussion accordingly.

2nd Editorial Decision

29 March 2017

Thank you for the submission of your revised manuscript to EMBO Molecular Medicine. We have now received the enclosed reports from the reviewers that were asked to re-assess it. As you see, while reviewer 2 is now supportive, reviewer 1 is decidedly more reserved.

Reviewer 1 is not satisfied that your revision addresses his/her main concern that your approach is informative regarding the properties of growing cancers and therefore that ultimately the conclusions are not adequately supported by the data.

After further discussion, reviewer 1 reiterated his/her position but also appreciation for an interesting, technically well-performed set of experiments. The reviewer also acknowledges that s/he has a different interpretation of your data. Ultimately however, s/he would not be opposed to publication of your manuscript, provided the findings were better contextualized and the limitations more clearly stated. I agree and also suggest that the spirit of reviewer 2's comments would also be best served by such a revision.

I am pleased to inform you that we will be able to accept your manuscript, provided you carefully consider and act upon the above. Please make sure the changes are highlighted in the manuscript. Finally, please also note the following pending final amendments:

- 1) We note that Fig S2 is called out in the manuscript before Fig. S1. Please reorganize in chronological order. We also note that there are no callouts for Tables S5, S6.
- 2) As per our Author Guidelines, the description of all reported data that includes statistical testing must state the name of the statistical test used to generate error bars and P values, the number (n) of independent experiments underlying each data point (not replicate measures of one sample), and the actual P value for each test (not merely 'significant' or 'P < 0.05').
- 3) The manuscript must include a statement in the Materials and Methods identifying the institutional and/or licensing committee approving the experiments, including any relevant details (like how many animals were used, of which gender, at what age, which strains, if genetically modified, on which background, housing details, etc). We encourage authors to follow the ARRIVE guidelines for reporting studies involving animals. Please see the EQUATOR website for details: <http://www.equator-network.org/reporting-guidelines/improving-bioscience-research-reporting-the-arrive-guidelines-for-reporting-animal-research/>. Please make sure that ALL the above details are reported, including in the checklist.
- 4) We encourage the publication of source data, with the aim of making primary data more accessible and transparent to the reader. Would you be willing to provide a PDF file per figure that contains the original, uncropped and unprocessed scans of all or at least the key gels used in the manuscript and/or source data sets for relevant graphs? The files should be labeled with the appropriate figure/panel number, and in the case of gels, should have molecular weight markers; further annotation may be useful but is not essential. The files will be published online with the article as supplementary "Source Data" files. If you have any questions regarding this just contact me.
- 5) We encourage the provision of striking image or visual abstract to illustrate your article. If you do, please provide a jpeg file 550 px-wide x 400-px high.

Please submit your revised manuscript within two weeks. I look forward to seeing a revised form of your manuscript as soon as possible.

***** Reviewer's comments *****

Referee #1 (Comments on Novelty/Model System):

The model system solely allows for the study of tumor initiation, and is not informative regarding properties of growing cancers. The conclusions of the authors are not supported by the data.

Referee #1 (Remarks):

As I also indicated in my initial review in my view there is an important conceptual problem with the model. All of the effects that are reported are potential artefacts of the engraftment phase of the assay and not of the features of tumor growth. Therefore, the conclusions as drawn by the authors, are not supported by the data.

The reply of the authors to my concerns is not very convincing, and does not present any new insights. Indeed, ten years ago the tumor initiation assay was accepted to be related to stem cells. In 2017 better assays are required to make claims about potential distinct types of stem cells within malignancies.

Unfortunately, no new data could be presented on natural engraftment, i.e. metastasis formation, as was also requested by the other reviewer.

Referee #2 (Remarks):

Authors argue that pancreatic model systems that they use do not generate metastasis, which precludes performing the experiments that I suggested. Instead, authors have discussed the limitations of the system in this revised version. My impression is that the work is relevant and should be published. Despite the technical caveats and potential artifacts that we discussed with the other reviewer (and that I subscribe), it is important to consider that similar transplantation experiments have been widely used to assess the hierarchical organization of other tumor types. In the case of CRC, authors applied the same lentiviral marking followed by serial transplantation. They, observed, however, that CRC is hierarchically organized, which argues against the possibility that their conclusion in pancreas cancer about clonal succession may simply represent a byproduct of the transplantation assay.

2nd Revision - authors' response

11 April 2017

Referee #1 (Comments on Novelty/Model System):

The model system solely allows for the study of tumor initiation, and is not informative regarding properties of growing cancers. The conclusions of the authors are not supported by the data.

Xenotransplantation of human cancer cells into severely immune deficient mice is widely used due to the lack of alternate experimental *in vivo* models for human cancer. We agree that it cannot be excluded that disruption of the original tumor's architecture, the transplantation procedure and the xenogenic environment may influence the behavior of the assayed tumor cell population. However, these assays have been adopted to successfully describe the presence and key properties, i.e. self-renewal and long-term activity, of tumor initiating cells in a wide range of human cancer tissue. We adapted these well-established approaches within our study and unexpectedly demonstrate that PDAC serial transplantation in this model unlike colon cancer does not require stably self-renewing tumor stem cells within a fixed malignant stem cell compartment. In the revised version of the manuscript, we have expanded the discussion of the limitations of the experimental model used and clearly contextualized the obtained results.

Referee #1 (Remarks):

As I also indicated in my initial review in my view there is an important conceptual problem with the model. All of the effects that are reported are potential artefacts of the engraftment phase of the assay and not of the features of tumor growth. Therefore, the conclusions as drawn by the authors, are not supported by the data.

The reply of the authors to my concerns is not very convincing, and does not present any new insights. Indeed, ten years ago the tumor initiation assay was accepted to be related to stem cells. In 2017 better assays are required to make claims about potential distinct types of stem cells within malignancies. Unfortunately, no new data could be presented on natural engraftment, i.e. metastasis formation, as was also requested by the other reviewer.

Indeed, continuous tumor growth cannot be assayed in serial xenotransplantation, as tumor explantation and dissociation before xenotransplantation are required. Therefore, it cannot be ruled out that the procedure of xenotransplantation per se impacts the clonal dynamics observed. However, we would like to emphasize that by using the same methodology we have previously demonstrated a hierarchical organization of the TIC compartment in colorectal cancer (Dieter et al., 2011), clearly establishing that the experimental model used is permissive for fundamentally different clonal dynamics of patient samples from different malignant diseases. Moreover, within the current study, we detected very similar clonal dynamics of cultured PDAC cells *in vitro* as we observed in serial xenografts *in vivo*, arguing against a dominant effect of xenogeneic engraftment on clonal TIC dynamics. Within the new version of the manuscript, we have thoroughly discussed the potential limitations of the model system in the introduction and discussion sections. Moreover, we have modified the title and text throughout the manuscript to more clearly indicate that the data presented are generated in the context of serial xenotransplantation.

Referee #2 (Remarks):

Authors argue that pancreatic model systems that they use do not generate metastasis, which precludes performing the experiments that I suggested. Instead, authors have discussed the limitations of the system in this revised version. My impression is that the work is relevant and should be published. Despite the technical caveats and potential artifacts

that we discussed with the other reviewer (and that I subscribe), it is important to consider that similar transplantation experiments have been widely used to assess the hierarchical organization of other tumor types. In the case of CRC, authors applied the same lentiviral marking followed by serial transplantation. They, observed, however, that CRC is hierarchically organized, which argues against the possibility that their conclusion in pancreas cancer about clonal succession may simply represent a byproduct of the transplantation assay.

We thank the reviewer for his positive comments.

Corresponding Author Name: Hanno Glimm

Manuscript Number: EMM-2016-07354 -V3